# Keratinases from *Streptomyces netropsis* and *Bacillus subtilis* and Their Potential Use in the Chicken Feather Degrading

Ali Abdelmoteleb [1], Daniel Gonzalez-Mendoza [2,*], Olivia Tzintzun-Camacho [2], Onecimo Grimaldo-Juárez [2], Vianey Mendez-Trujillo [3], Carlos Moreno-Cruz [2], Carlos Ceceña-Duran [2] and Ahmed F. Roumia [4]

1   Botany Department, Faculty of Agriculture, Menoufia University, Shibin El-Kom 32514, Egypt
2   Institute of Agricultural Sciences, Autonomous University of Baja California (ICA-UABC), Highway to Delta s/n C.P, Ejido Nuevo León 21705, Baja California, Mexico
3   Faculty of Medicine, Autonomous University of Baja California, Dr. Humberto Torres Sanginés S/N, Centro Cívico, Mexicali 21000, Baja California, Mexico
4   Department of Agricultural Biochemistry, Faculty of Agriculture, Menoufia University, Shibin El-Kom 32514, Egypt
*   Correspondence: danielg@uabc.edu.mx

**Abstract:** Feathers are the most prevalent agricultural waste generated by chicken farms, polluting the environment and wasting protein resources as a result of the accumulation of large amounts of feathers. Therefore, keratinase-producing microorganisms represent a promising potential technique for the degradation of feather waste. *Streptomyces netropsis* A-ICA and *Bacillus subtilis* ALICA, previously isolated from the rhizosphere of desert plants (*Larrea tridentata* and *Prosopis juliflora*) respectively, were assessed for their feather-degradation ability. Keratinase activity was optimized using various parameters, including incubation time, pH, temperature, and feather concentration. The maximum keratinase activity of *S. netropsis* A-ICA and *B. subtilis* ALICA (113.6 ± 5.1 and 135.6 ± 4.1 U/mL) was obtained at the 5th and 3rd day of incubation with initial pH of 7.0 and 7.5 at 25 and 30 °C, and 1% (*w/v*) of chicken feather, respectively. Under the optimized conditions, the concentration of soluble protein in the feather hydrolysate reached 423.3 ± 25 and 565.3 ± 7.7 μg/mL, with feathers weight loss of 84 ± 2 and 86± 1.5% by *S. netropsis* A-ICA and *B. subtilis* ALICA, respectively. The highest disulphide bond reductase activity reached 10.7 ± 0.4 and 10.96 ± 1.1 U/mL, after five and three days of inoculation with *S. netropsis* A-ICA and *B. subtilis* ALICA, respectively. Furthermore, the antioxidant activity of feather protein hydrolysate obtained *by S. netropsis* A-ICA and *B. subtilis* ALICA was evaluated using DPPH radical-scavenging activity, which exhibited a significant antioxidant potential with an $IC_{50}$ value of 0.8 and 0.6 mg/mL. The 3D models of detected keratinases in both strains showed high similarity with subtilisin family. Further, the docking results clarified the importance of GSG and VVVFTP domains in *B. subtilis* and beta-keratin, respectively. The present study revealed the keratinolytic potential of *S. netropsis* A-ICA and *B. subtilis* ALICA in chicken feather degradation, which have potential application value and may be exploited as supplementary protein and antioxidant in animal feed formulations.

**Keywords:** waste pollution; biodegradation; fermentation; keratinolytic; antioxidants

## 1. Introduction

Feathers are the principal waste by-product of the poultry industry, and annually millions of tons of feather waste are produced globally [1]. Chicken feathers contain approximately 90% crude protein, amino acids, high-value elements, growth factors, and vitamins [2]. A small proportion of these feathers is used in insulating materials, and down products, while the majority are burned or discarded as waste [2]. Due to the presence of keratin as the main constituent, chicken feathers exhibit extreme rigidity and resistance to chemical, physical, and biological agents as well as resistance to proteolysis by common proteases, such as pepsin, trypsin, and papain, thus slowing the process of feather

degradation in the nature [3,4]. Chicken feathers mainly consist of β-keratin (over 90%), which is a highly stable and insoluble fibrous structural protein that represents a valuable source of amino acids, which are exploited in several applications such as animal feed, and fertilizers [5]. Keratin is an insoluble protein containing hydrogen and disulphide bonds that make it stable, rigid, and difficult to degrade [6]. Disulphide bonds can form crosslinks between protein peptide chains, resulting in a dense polymeric structure in combination with hydrogen bonding and hydrophobic forces, and this gives keratin quite good stability, rigidity, and strong mechanical resistance [2].

Traditional disposal of keratinous waste by burning has a harmful effect on environment, as well as is restricted or prohibited in several countries, thus, the developing of alternative techniques for this waste disposal is a quite important [7]. The conventional hydrothermal and chemical (acid and alkali) processes currently utilized in the industry of feather meal are energy-intensive and produce low soluble and a poorly digestible protein. The high pressure and temperatures used in these methods lead to product damage and release significant amounts of waste gases, such as ammonia and sulfur dioxide, making them polluting and unsustainable [2,8]. Furthermore, these techniques result in the loss of several thermo-sensitive amino acids, e.g., lysine, methionine, and tryptophan, as well as the addition of non-nutritive amino acids, such as lysinoalanine and lanthionine [9]. Therefore, the economic degradation of feathers is of great interest among researchers. Among the various techniques, the biodegradation of keratin by microbial keratinolytic proteases is viable as an effective alternative for physical and chemical approaches, due to their rapid growth, availability, high yield, cost, less space requirement, and sustainability [10]. The biodegradation of the keratinous substances by microorganisms is an eco-friendly and inexpensive approach for the effective processing of keratinous wastes that entails environmental safety, waste management, and resource generation, e.g., of peptides and amino acids [11]. Additionally, the enzymatic hydrolysis of feathers is able to protect the majority of the amino acids that are typically destroyed by physical and chemical treatments and improve their nutritional properties [5]. Several microorganisms, including fungi, Actinomycetes, and bacteria, can degrade the keratinous substrates [12]. Among bacteria, *Bacillus* spp., e.g., *B. subtilis*, *B. cereus*, and *B. licheniformis*, among others are effective for degrading feathers due to their fast growth rate and safety [13]. *Streptomyces* sp. is one of the most promising producers of extracellular keratinolytic enzymes [14].

Keratinases are proteolytic enzymes that can hydrolyze keratin and are exploited in various industries, such as detergent and leather industries, and in animal feed additives [5]. Keratinase activity involves disulphide reductase and polypeptide hydrolase. The main mechanisms of keratin degradation are denaturation, hydrolysis, and transamination. Firstly, the strongly bound structure is changed to its degenerative state by the reduction of the disulphide link in keratin from cystine (-S-S-) to cysteine (-SH). The subsequent step is keratin hydrolysis into polypeptides, oligopeptides, and free amino acids. Finally, the keratin is completely degraded into free amino acids and sulphides [6]. Thus, the biodegradation of chicken feathers using keratinolytic microorganisms presented a practical alternative for the efficient bioconversion of keratinous wastes into animal feed [15]. Natural antioxidants are among the many dietary components that are crucial for animal health and protect the body against free radicals. Feather wastes are an attractive source of proteins and antioxidant peptides [2,15].

Due to the promising applications of keratinase in various industries, it became crucial to investigate various conditions for bacterial diversity with the potential to develop novel keratinases [16]. In the present study, we have used two strains previously identified as *Streptomyces netropsis* A-ICA and *Bacillus subtilis* ALICA, which have been isolated from the rhizosphere of *Larrea tridentata* and *Prosopis juliflora* respectively [17,18]. Therefore, this study was conducted to evaluate the keratinolytic potentials of these strains using chicken feathers as a sole carbon and nitrogen source. The fermentation conditions were optimized for maximum keratinase production. In addition, the potential of keratinase-producing strains to degrade chicken feathers and produce antioxidant hydrolysate was assessed

under optimized conditions. Finally, bioinformatic analysis, including proteases detection, 3d modelling of potential keratinases, as well as protein–protein interaction simulation, have been performed to define the hot spots either in beta-keratin or detected keratinases.

## 2. Materials and Methods

### 2.1. Microorganisms

*S. netropsis* strain A-ICA and *B. subtilis* strain ALICA were previously isolated as plant growth promoting and antagonistic strains from rhizosphere of *L. tridentata* and *P. juliflora* respectively, from the desert region of Baja California, Mexico. These strains were identified based on 16S rDNA sequences and submitted to the GenBank with accession number MN535765 and KX137176 respectively [17,18].

### 2.2. Poultry Feathers and Chemical Materials Used in This Study

Chicken feathers were collected from the chicken slaughtering market and thoroughly cleaned by washing with tap water, then with distilled water. The clean feathers were dried at 55 °C for 2 days, and then feathers were cut into pieces of 2–3 cm and stored until used.

The chemical materials used in this study include casein, potato dextrose media, $CaCl_2$, $MgSO_4 \cdot 7\,H_2O$, $K_2HPO_4$, $KH_2PO_4$, $NaH_2PO_4$, $Na_2HPO_4$, Trichloroacetic acid, $Na_2CO_3$, Folin–Ciocalteu's phenol reagent, tyrosine, NaOH, 5,5'-dithiobis-(2-nitrobenzoic acid) (DTNB), 2,2-diphenyl-1-picrylhydrazyl (DPPH), and methanol.

### 2.3. Primary Detecting of Keratinase Activity

The keratinolytic activity of both *S. netropsis* strain A-ICA and *B. subtilis* strain ALICA was qualitatively tested on casein agar plates containing 2% (*w/v*) casein. *S. netropsis* A-ICA was grown in 50 mL potato dextrose broth at 30 °C for one week at 120 rpm. Hence, 5 µL of spore suspension was spot inoculated in the center of casein plates, while *B. subtilis* strain ALICA was directly streaked on casein agar plates. The plates were incubated at 33 ± 2 °C for 72 h, and the keratinolytic activity was observed by the appearance of a clear hydrolysis zone around the colony after the incubation time [13].

### 2.4. Optimization of Culture Conditions and Quantitative Assay of Keratinase Activity

Culture conditions: 2 mL of *S. netropsis* A-ICA ($10^6$ CFU/mL) and *B. subtilis* ALICA ($10^8$ CFU/mL) were inoculated separately in 500 mL Erlenmeyer flask containing 200 mL of keratinase production media (Components g/L): $CaCl_2$ 0.22, $MgSO_4 \cdot 7\,H_2O$ 0.2, $K_2HPO_4$ 0.3, $KH_2PO_4$ 0.4, poultry feather 5) [19]. Several culture conditions were tuned to get the most keratinase possible from *S. netropsis* A-ICA and *B. subtilis* ALICA as follows: (a) Incubation period: To assess the effect of incubation period on keratinase production, the inoculated media (pH = 7) was incubated at 30 °C and 120 rpm for 10 days. Every 24 h, 2-mL samples were taken and the microbial growth removed by centrifugation of the culture at 12,000 rpm and 4 °C for 8 min, and the supernatant was used as a crude enzyme solution for keratinase assay. (b) pH: To check the impact of initial pH on keratinase activity, *S. netropsis* A-ICA and *B. subtilis* ALICA were inoculated in basal salt medium containing 1% feathers with different pH 6.0, 6.5, 7, 7.5, 8, 8.5, and 9.0 for 6 days at 30 °C and 120 rpm. Keratinase activity was estimated using cell free supernatant. (c) Temperature: For determining the optimum temperature for keratinase production and feather degradation by *S. netropsis* A-ICA and *B. subtilis* ALICA, these strains were grown in basal medium adjusted with 1% feathers, pH = 7, for 6 days at diverse temperatures ranging from 20 to 40 °C with 5 °C interval. After incubation, the supernatant was used for keratinase activity estimation; (d) feather concentration: The effect of feather concentration on keratinase production was studied by the inoculation of microorganisms in the basal medium prepared with different feather concentrations (0.5, 1, 1.5, 2, and 2.5%). (e) Keratinase assay: Casein solution 2% (*w/v*) was prepared in 50 mM phosphate buffer (pH 7) and used as substrate. Equal volumes (1 mL) of both crude enzyme and substrate solution were mixed and incubated for 20 min at 40 °C in a water bath. Then, 2 mL of 0.4 M Trichloroacetic acid was added to the mixture

for terminating the reaction, and kept at room temperature for 30 min. The mixture was centrifuged for 10 min at 12,000 rpm and 4 °C to remove the precipitated proteins, and finally 0.5 mL of supernatant, 2.5 mL of 0.4 M $Na_2CO_3$ and 0.75 mL of Folin–Ciocalteu's phenol reagent: water (1:3 *v/v*) were mixed and incubated under dark condition for 30 min at room temperature. The absorbance of the developed color was measured using UV–vis spectroscopy (Thermo Scientific BioMate 3 Spectrophotometer, Waltham, MA, USA) at 660 nm, and the optical density of the samples was compared against tyrosine standard curve. The required amount of the enzyme to liberate 1 µg of tyrosine per minute under the standard assay conditions was defined as one unit of keratinase activity [13].

*2.5. Feathers Degradation under Optimized Fermenting Conditions*

The feather degradation experiments were verified under the optimized conditions in 2-L Erlenmeyer flasks containing 1 L of the basal media supplemented by 1% (*w/v*) of chicken feather waste. The pH media was adjusted to 7 and 7.5 for *S. netropsis* A-ICA and *B. subtilis* ALICA, which were inoculated separately and incubated at 25 and 30 °C for 10 days, respectively.

At the end of incubation period, the feathers degradation rate (% of substrate weight loss) was calculated using the difference between the dry weight of pre- and post-decomposition [2]. Unhydrolyzed feathers were removed from the hydrolysate by passing it through Whatman No. 1 filter paper. The feathers were then thoroughly washed with deionized water to remove any soluble materials and microorganisms, and then dried for 24 h at 60 °C. The formula used to determine the feather degradation rate (DR) is as follows:

$$(DR) = ((A - B)/A) \times 100$$

where A represents the dry weight of the feathers prior to decomposition and B represents the dry weight of the feathers following decomposition. Total protein of the supernatant has been estimated after 1, 3, 5, 7, and 9 days of incubation according to Bradford method [20] and using bovine serum albumin standard curve.

*2.6. Disulphide Bond Reductase Activity and Keratinase Activity under Optimized Conditions*

The reductase activity of disulphide bond was measured spectrophotometrically at 412 nm in the supernatant by evaluating the yellow-colored sulphide generated upon reduction of 5,5′-dithiobis-(2-nitrobenzoic acid) (DTNB) to confirm the occurrence of thiolysis [21]. Briefly, 1 mL of DTNB was mixed with 0.5 mL of supernatant, and the mixture was left at room temperature for 5 min to develop the color. The absorbance of yellow 2-nitro-5-thiobenzoic acid (TNB) generated from the reduction of DTNB was measured by a spectrophotometer at 412 nm. Under the given conditions, 0.01 increase in absorbance was expressed as one unit of disulphide bond reductase activity [13]. Keratinase assay was measured as described above.

*2.7. Antioxidant Activity of Chicken Feather Hydrolysate*

The 2,2-diphenyl-1-picrylhydrazyl (DPPH) radical-scavenging activity of the feather hydrolysate was estimated. The mixture of 1.9 mL of DPPH in methanol and 0.1 mL of hydrolysate sample at several concentrations (0.0, 0.1, 0.5, 1.5, 2.0, and 2.5 mg/mL) was incubated in the darkness at room temperature for 60 min. The scavenging activity of DPPH (absorbance decrease) was measured by UV–visible spectrophotometer at 517 nm [15]. The % inhibition of the DPPH radical (scavenging activity) was determined using the following formula:

$$\% \text{ of scavenging activity} = ((\text{Control OD} - \text{Sample OD})/(\text{Control OD})) \times 100$$

The graph displaying the percentage of inhibition against the concentration of the feather hydrolysate was used to calculate the sample concentration that provides 50% inhibition ($IC_{50}$).

### 2.8. Bioinformatics Analysis for Keratinases in Studied Strains

The complete genomes of *S. netropsis* and *B. subtilis* (strain 168) were downloaded from Uniprot (Release 2022_04) [22]. Further, 14 different protease families were retrieved from MEROPS database (Release 12.1) [23]. The Peptidase_S8 (PF00082.24), the characteristic profile hidden Markov model (pHMM) of keratinases, and the 3D structure of β-keratin (PF02422) were downloaded from InterPro [24].

To detect all the existent proteases of studied strains, a BLAST search was performed for the complete genomes of *S. netropsis* and *B. subtilis* (strain 168) against the 14 families of MEROPS database. An E-value of 0.0001 was set as the cutoff value. Moreover, we assigned the existed keratinases in both strains using HMMER package [25] by running hmmsearch of the PF00082.24 against the two studied genomes. Finally, the redundancy between the detected keratinases was reduced to 30% using MMseqs2 [26].

SWISS-MODEL server [27] was used to predict the 3D structure of the keratinase hits in both strains. For every hit, a protein–protein interaction was executed with the 3D model of β-keratin (PF02422) via ClusPro server [28].

### 2.9. Statistical Analysis

One-way analysis of variance (ANOVA) was used to conduct statistical analyses of all data using PASW statistics 21.0. (IBM Inc., Chicago, IL, USA). Tukey's honestly significant difference (HSD) was used to compare the different means of traits for different treatments, and all results were considered significant at $p \leq 0.05$. All data were expressed as mean of triplicates $\pm$ standard deviation.

## 3. Results and Discussion

### 3.1. Screening of Keratinase Activity

In this study, *S. netropsis* A-ICA and *B. subtilis* ALICA previously isolated from the rhizosphere of *L. tridentata* and *P. juliflora* respectively were subjected for preliminary screening on casein agar plates. Keratinase activity was indicated in both strains by observing the clear zone formation around their colonies (Figure 1).

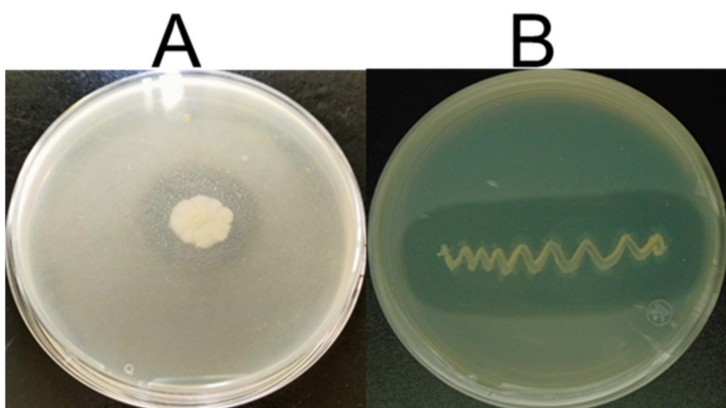

**Figure 1.** Preliminary screening of keratinase production by two strains (**A**) *S. netropsis* A-ICA; (**B**) *B. subtilis* ALICA exhibiting zone of hydrolysis around the colonies on the casein plate.

Both strains showed a hydrolysis clear zone of 14 and 17 mm, respectively. Feathers are a keratinous waste with high protein content, which can be utilized in a variety of applications using developing biotechnological strategies. However, these strategies face significant difficulties as keratin is difficult to digest and demonstrates considerable resistance to physical and chemical treatments and typical proteases enzymes, although keratin can be degrading by the proteolytic enzymes known as keratinases [5]. Keratinase producing microorganisms can be used as alternative and cheap method to get rid of feather wastes by converting into peptides and amino acids which can use as low-cost supplements for foodstuff and animal feed [29,30]. The microbial keratinases are of special

interest in the context of biotechnological processes due to their role to degrade feather wastes. Keratinase production from soil bacteria has been documented for industrial uses. Most Gram-positive bacteria, particularly soil microorganisms Bacilli and Actinomycetes, produce keratinase [10].

### 3.2. Optimization of Culture Condition for Keratinase Activity

The strains *S. netropsis* A-ICA and *B. subtilis* ALICA exhibited remarkable feather degradation in the broth medium amended with feathers as a sole nitrogen and carbon source. The extracellular keratinase production by these strains was evaluated daily during 10 days of incubation. The results of both strains revealed that keratinase activity was low at the first of incubation period. Then, the enzyme activity significantly increased until reaching the peak of keratinolytic activity. After that, the enzyme activity was gradually decreased with more incubation times. The data presented in Figure 2A illustrate that the highest significant ($p < 0.05$) keratinase activity of $36.8 \pm 3.1$ and $50.56 \pm 0.7$ U/mL was observed at the 5th and 3rd day of incubation by *S. netropsis* A-ICA and *B. subtilis* ALICA respectively. The highest keratinase activity was occurred at the exponentially microbial growth phase, indicating that the tested microorganisms secreted keratinase enzyme as principal metabolite [29]. Similar results were reported by several researchers [2,13,16]. The metabolites released by the microorganisms and the byproducts of enzyme activities would be a logical explanation for the observed pattern. The organic acid may be one of the early-stage metabolites released that increased the acidity of the media. In later stages, alkaline-based metabolites, including those connected to nitrogen, would have prevailed. Deamination of amino acids resulting from keratin degradation is the primary cause of alkalization [16].

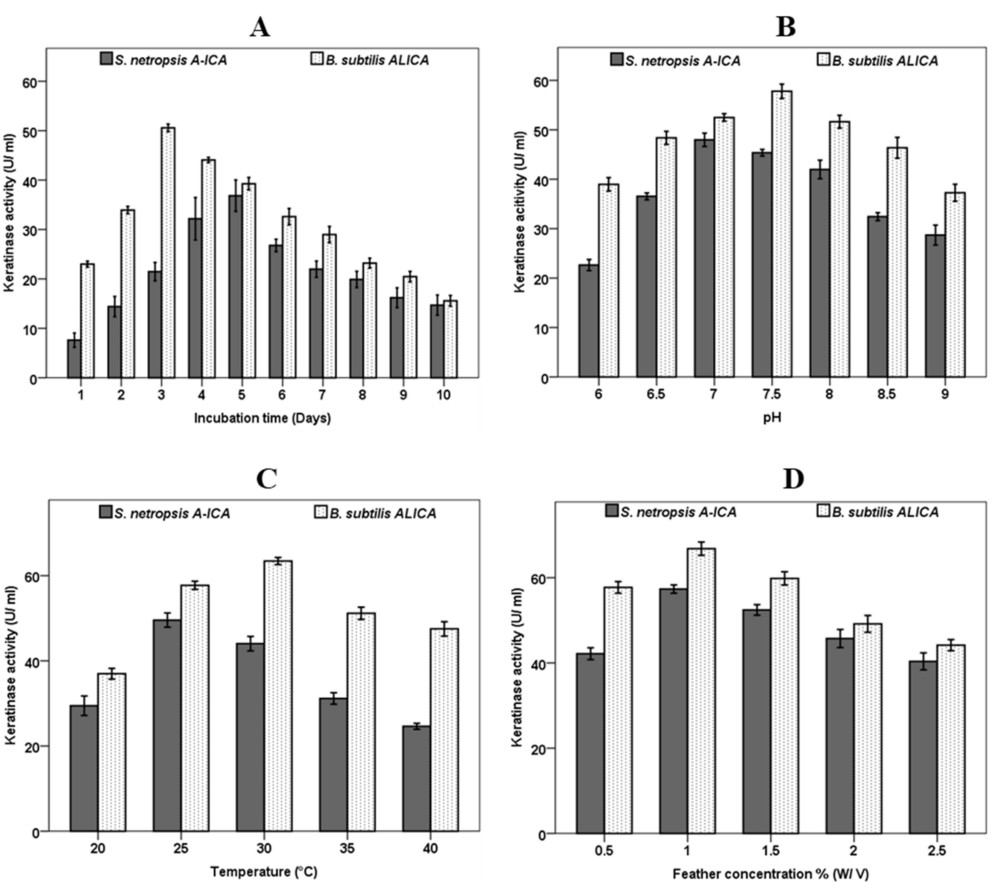

**Figure 2.** Effect of incubation time (**A**), pH (**B**), Temperature (**C**) and Feathers concentration (**D**) on Keratinase activity produced by *S. netropsis* A-ICA and *B. subtilis* ALICA.

### 3.3. Effect of Culture pH and Fermentation Temperature on Keratinase Activity

The effect of initial pH of culture media on keratinase production was observed at different pH levels ranging from 6 to 9. Generally, the keratinase production was significantly impacted by the initial pH in both microorganisms. The results presented in Figure 2B revealed that the highest keratinase production of $47.9 \pm 1.3$ and $57.8 \pm 1.4$ U/mL was observed at pH 7 and 7.5 by *S. netropsis* A-ICA and *B. subtilis* ALICA, respectively. Similarly, it was reported that the optimum pH for keratinase production by *B. tropicus* Gxun-17 was neutral to weakly alkaline [13]. Moreover, *S. coelicoflavus* showed maximum keratinase production at pH 8.0 [31]. Generally, the most suitable pH for keratinase production from Actinomycetes, fungi, and bacteria was found to neutral to alkaline range [29]. On the other hand, it has been observed that temperature has a remarkable impact on keratinase activity in both examined strains (Figure 2C). The variation of culture temperature exhibited that the keratinase activity was at its peak ($49.5 \pm 1.7$ and $63.4 \pm 0.8$ U/mL) at 25 and 30 °C in *S. netropsis* A-ICA and *B. subtilis* ALICA, respectively. Keratinase activity continuously reduced as fermentation temperature increased which suggests the mesophilic nature of *S. netropsis* A-ICA and *B. subtilis* ALICA.

In this context, keratinase enzyme was optimally released by *Bacillus* sp. FPF-1 at 25 °C, then the enzyme activity dropped continuously with increasing temperature [16]. Moreover, *S. coelicoflavus* exhibited maximum keratinase activity at 40 °C [31]. Numerous cultural conditions have an impact on feather deterioration. The efficiency of feather decomposition depends on the medium pH and fermentation temperature, which differ from strain to other. The optimal pH and temperature for degradation of chicken feathers by *S. netropsis* A-ICA and *B. subtilis* ALICA were neutral pH (pH 7.0 and 7.5) and moderate temperature (25 and 30 °C), respectively. At higher pH and temperature, feather degradation was slowed down. The capacity of these strains to degrade the feather at neutral pH and moderate temperature is considered a desirable and attractive outcome due to the possibility of several essential amino acids being destroyed during feather digestion at higher alkaline (pH > 9) levels [15].

### 3.4. Effect of Feather Concentration on Keratinase Activity

Among different feather concentrations, 1% (*w/v*) significantly increased keratinase production (Figure 2D). The maximum keratinase activity of $57.3 \pm 0.9$ and $66.8 \pm 1.5$ U/mL by *S. netropsis* A-ICA and *B. subtilis* ALICA, respectively was recorded at 1% (*w/v*) of chicken feather in the fermentation media. Then, keratinase activity was constantly decreased with further increase of feather concentration more than 1%. Similarly, *B. tropicus* Gxun-17 reached its peak of keratinase production at 15 g/L of feathers, then decreased at higher concentrations of feather [13]. This may be due to the fermentation broth's comparatively high viscosity, caused by high concentrations of feather, which affect the bacterial growth and the release of keratinase by reducing the dissolved oxygen supply to the fermentation system [15,16,32]. The higher feather concentration leads to an increase in viscosity and decrease in aeration in the fermentation medium, which impedes the microbial growth as well as retards feather degradation [15]. It has been reported that the majority of bacteria achieved their peak of keratinase activity at feather concentrations range of 5–20 g/L [13]. *S. coelicoflavus* produced the most keratinase when feather concentrations were 1% (*w/v*), and this production marginally decreased as feather concentrations increased [31].

### 3.5. Feathers Degradation under Optimized Condition

The concentration of soluble protein in the hydrolysate gradually increased throughout the course of the incubation period, reaching $423.3 \pm 25$ and $565.3 \pm 7.7$ µg/mL by *S. netropsis* A-ICA and *B. subtilis* ALICA respectively, after nine days of hydrolysis, (Figure 3A). Generally, with the prolongation of the fermentation time, the enzyme activities decreased, whilst the soluble protein concentration was continuing to increase. Total protein in the cell free broth was increased as time progressed, indicating keratin degradation. The high potential of microbial keratinases degrades keratinous biomass

quicker than the producing bacteria can utilize it, resulting in an increase the total protein concentration in the fermentation broth [16]. Our results are in accordance with previous reports [7,33]. On the other hand, the weight loss of feathers was measured after one, three, five, seven, and nine days of fermentation. The rate of feather weight loss significantly increased over time, reaching its peak of 84 ± 2 and 86± 1.5% for *S. netropsis* A-ICA and *B. subtilis* ALICA after seven and five days respectively, showing the complete degradation of feathers by both strains (Figure 3B). It was observed that the medium's total protein level remarkably increased in response to the degradation of the feather substrate, providing a measure of the degradation process's effectiveness. These outcomes are consistent with those attained by Kshetri et al. [15], who got 84% feather degradation in addition to an increase in soluble protein concentration. Similarly, the degradation rate of feathers by *Bacillus velezensis* NCIM 5802 was found to about 90% under optimized conditions [11]. The weight loss of the keratin substrate is the strongest evidence of the keratinolytic capacity of microorganisms [6].

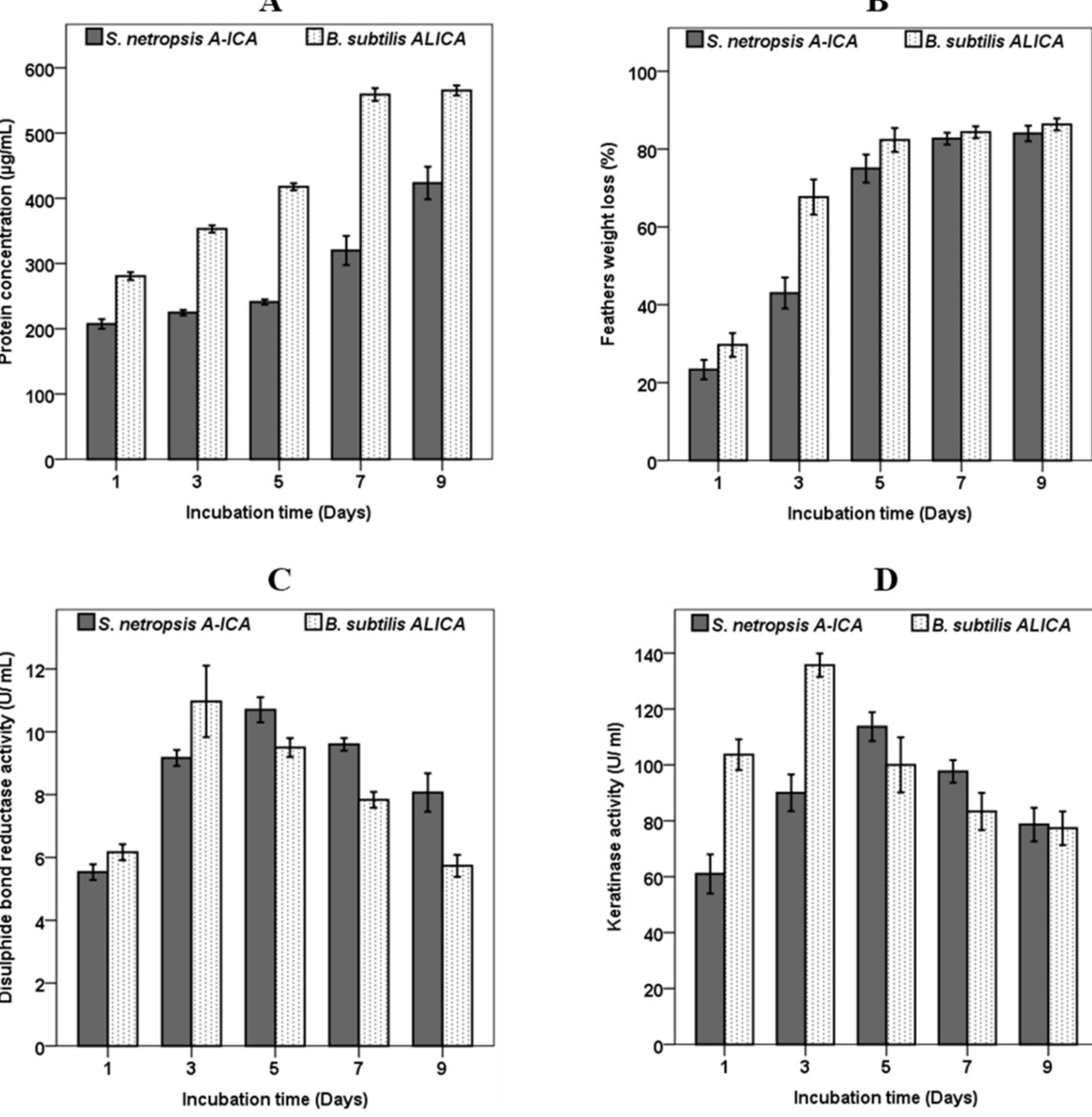

**Figure 3.** Total protein concentration (**A**), feather weight loss (**B**), and disulphide bond reductase activity (**C**), and keratinase activity (**D**) by *S. netropsis* A-ICA and *B. subtilis* ALICA separately, in the basal feather medium during 9 days of fermentation.

Disulphide Bond Reductase Activity and Keratinase Activity under Optimized Conditions

As illustrated in Figure 3C, disulphide bond reductase activity gradually increased and peaked at five and three days after inoculation with *S. netropsis* A-ICA and *B. subtilis* ALICA, with the highest disulphide bond reductase activity reaching $10.7 \pm 0.4$ and $1.96 \pm 1.1$ U/mL, respectively. Disulphide bond reductase activity declined after five and three days of fermentation by *S. netropsis* A-ICA and *B. subtilis* ALICA, respectively. Under the optimized conditions, the keratinase activity of *S. netropsis* A-ICA and *B. subtilis* ALICA reached $113.6 \pm 5.1$ and $135.6 \pm 4.1$ U/mL, which was 2.4- and 2.36-fold that obtained before optimizing conditions (47 and 57 U/mL) respectively (Figure 3D). There are four theories that explain the mechanisms of feather degradation, including mechanical pressure theory, biological membrane potential theory, thiolysis theory, and enzymolysis theory. However, the essence of each theory depends on the fracture of disulphide bonds [13]. In the present study, both *S. netropsis* A-ICA and *B. subtilis* ALICA produced a significant amount of keratinase and disulphide bond reductase, which simultaneously contribute to the process of feather degradation. It was found that disulphide bond reduction significantly facilitates the complete degradation of rigid keratin through structural modification, and this structural alteration makes keratin more susceptible to proteolytic hydrolysis [16]. The extraordinarily complicated structure of keratinous substrates, which contains a significant number of disulfide bonds, prevents traditional proteases from acting upon them. In contrast, the keratinase enzymes are able to easily degrade the keratin's complicated structure by reducing the number of disulphide bonds and breaking complex keratin structures [34]. The keratin degradation mechanism by microorganisms depends on two protein enzymes' cooperation. Here, disulfide reductase will decrease disulfide bonds and form denatured keratin (opened keratin structure), which will be degraded by protease enzymes into peptides and amino acids [6,34]. Although the activity of two enzymes produced by *B. tropicus* Gxun-17 decreased with the prolongation of fermentation time, the feather degradation rate was continuously increased, and this indicated that the two enzymes' impacts may not be the only action of the feather degradation mechanism [13]. There are a large variety of keratinolytic microbes that exist in nature and are capable of decreasing the disulfide bonds in complex and rigid keratin structures, rendering them susceptible to proteolytic destruction, thereby resolving the issue of excessive waste produced by keratinous wastes [34,35].

*3.6. Antioxidant Activity of Chicken Feather Hydrolysate*

DPPH radical scavenging activity was used to assess the antioxidant capacity of feather protein hydrolysate at different concentrations (0.1, 0.5, 1.0, 1.5, 2.0, and 2.5 mg/mL), and the results revealed that the inhibitory activity increases as hydrolysate concentration increased. As shown in Figure 4, feather hydrolysate exhibited remarkable radical scavenging activity with an $IC_{50}$ value of 0.8 and 0.6 mg/mL using *S. netropsis* A-ICA and *B. subtilis* ALICA, respectively. The feather hydrolysate with *S. netropsis* A-ICA and *B. subtilis* ALICA exhibited lower $IC_{50}$ values (indicating higher antioxidant abilities) than chemically prepared keratin hydrolysate [36]. It was observed that the antioxidant activity of feather hydrolysate obtained using *S. netropsis* A-ICA ($IC_{50}$ = 0.8 mg/mL) and *B. subtilis* ALICA ($IC_{50}$ = 0.6 mg/mL) is higher than feather hydrolysate prepared using *Streptomyces* sp. MAB18 ($IC_{50}$ = 78 mg/mL) [37]. Moreover, antioxidants not only increase the nutritional value of animal feed, but also protect it against deterioration [15].

*3.7. Bioinformatics Analysis for Keratinases in Studied Strains*

Keratinases are formerly ordered with a full four-digit EC number, and multiple keratinase sequences are accessible in the NCBI database. The common sequenced keratinases are under one of 14 different protease families, namely M3, M4, M14, M16, M28, M32, M36, M38, M55, S1, S8, S9, S10, and S16. The families of M4, M16, M36, S1, S8, and S16 act as endo-attack proteases, while M14, M28, M38, M55, S9, and S10 families work as exo-attack proteases, while M3 and M32 behave as oligopeptides [4]. Figures 5 and 6 depict

the detected hits of proteases in *B. subtilis* (strain 168) and *S. netropsis*, respectively. While the former has 12,679 hits in 67 unique proteins, the latter contains 19,660 different hits in 120 in distinctive proteins. Both strains contain seven different keratinase hits that belong to S8 family. Members of this family are endopeptidases with a catalytic triad in the order Asp, His, and Ser in their sequences. A standard S8 protein structure encompasses three layers with a seven-stranded β sheet squeezed by two layers of helices [23]. Molecular docking enabled us to predict the interactions between beta-keratin and keratinases from both strains and cause them to be in a conjugated system. A molecular docking technique was employed in this work to support the results of the analysis. Figure 7 reveals the possible interaction between beta-keratin and the predicted 3D models of eight different keratinases (two of *B. subtilis* (strain 168) and six of *S. netropsis*). Among these interactions, some interesting findings have been found. For example, the GSG interacted with the domain in both keratinases of *B. subtilis* where the serine acts as the nucleophilic amino acid at the keratinase active sites [38]. Moreover, the studied keratinases bind to their substrate via active sites which are distributed commonly along the first and second loops. In addition, beta-keratin interacts with the aforementioned keratinases with a common binding domain (VVVFTP) located at its second loop.

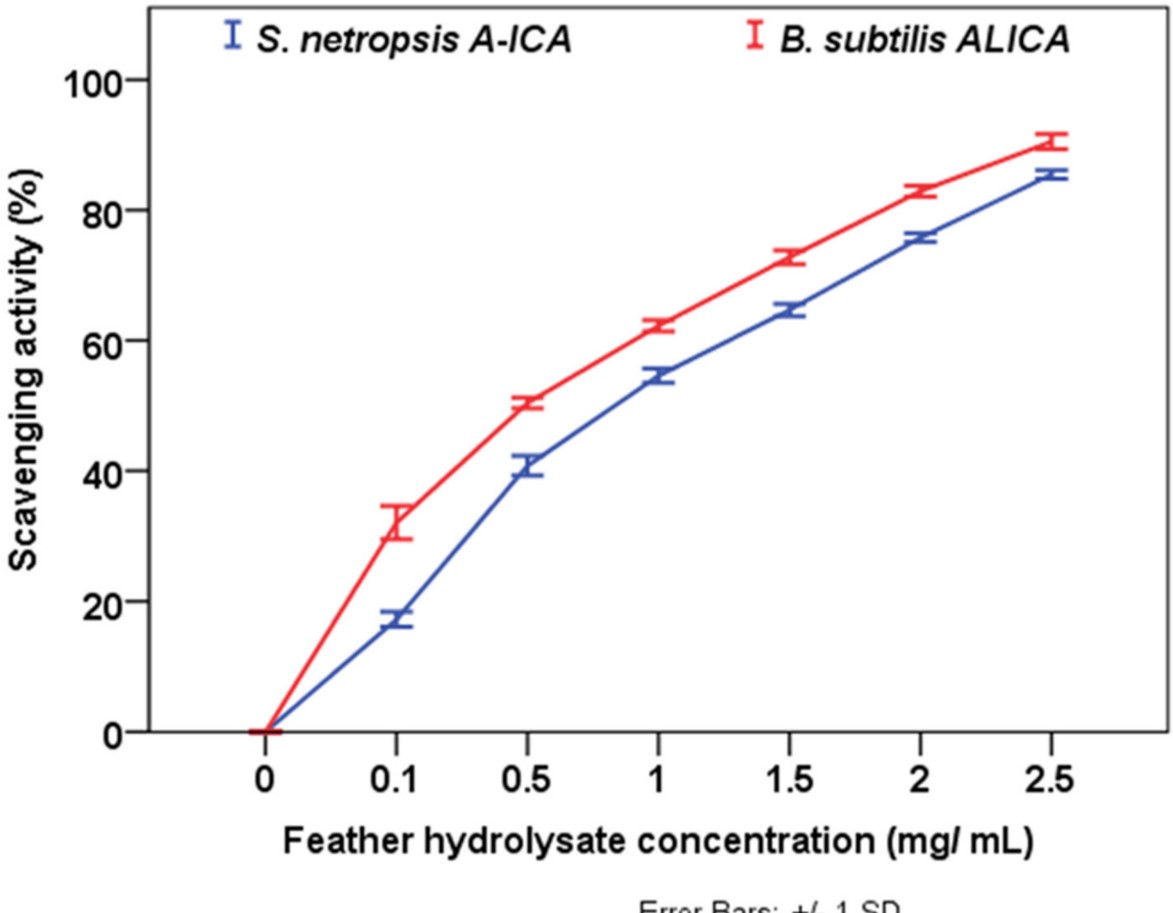

**Figure 4.** Inhibition percentage of DPPH antioxidant activity of feather hydrolysates.

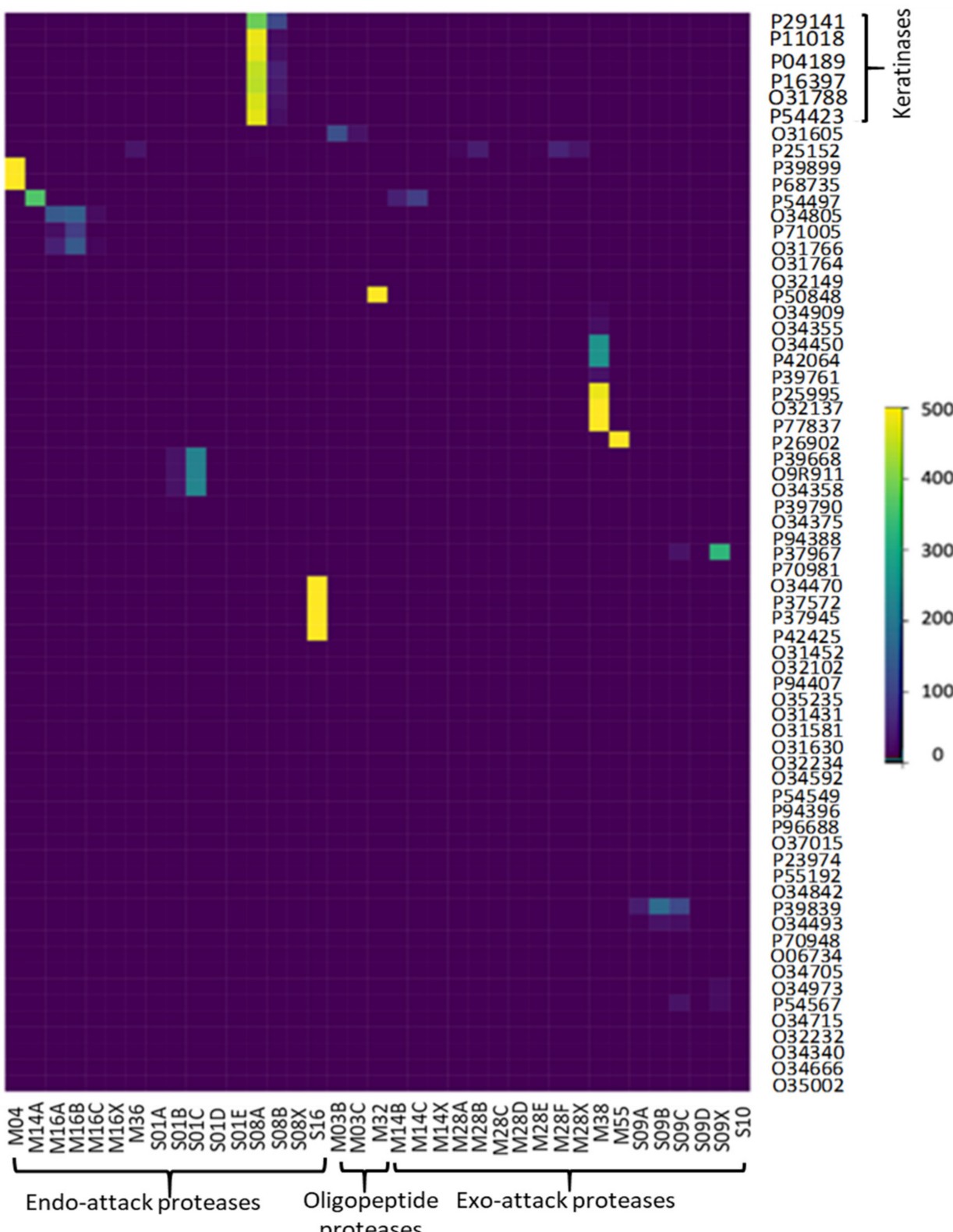

**Figure 5.** Detected proteases of *B. subtilis* (strain 168).

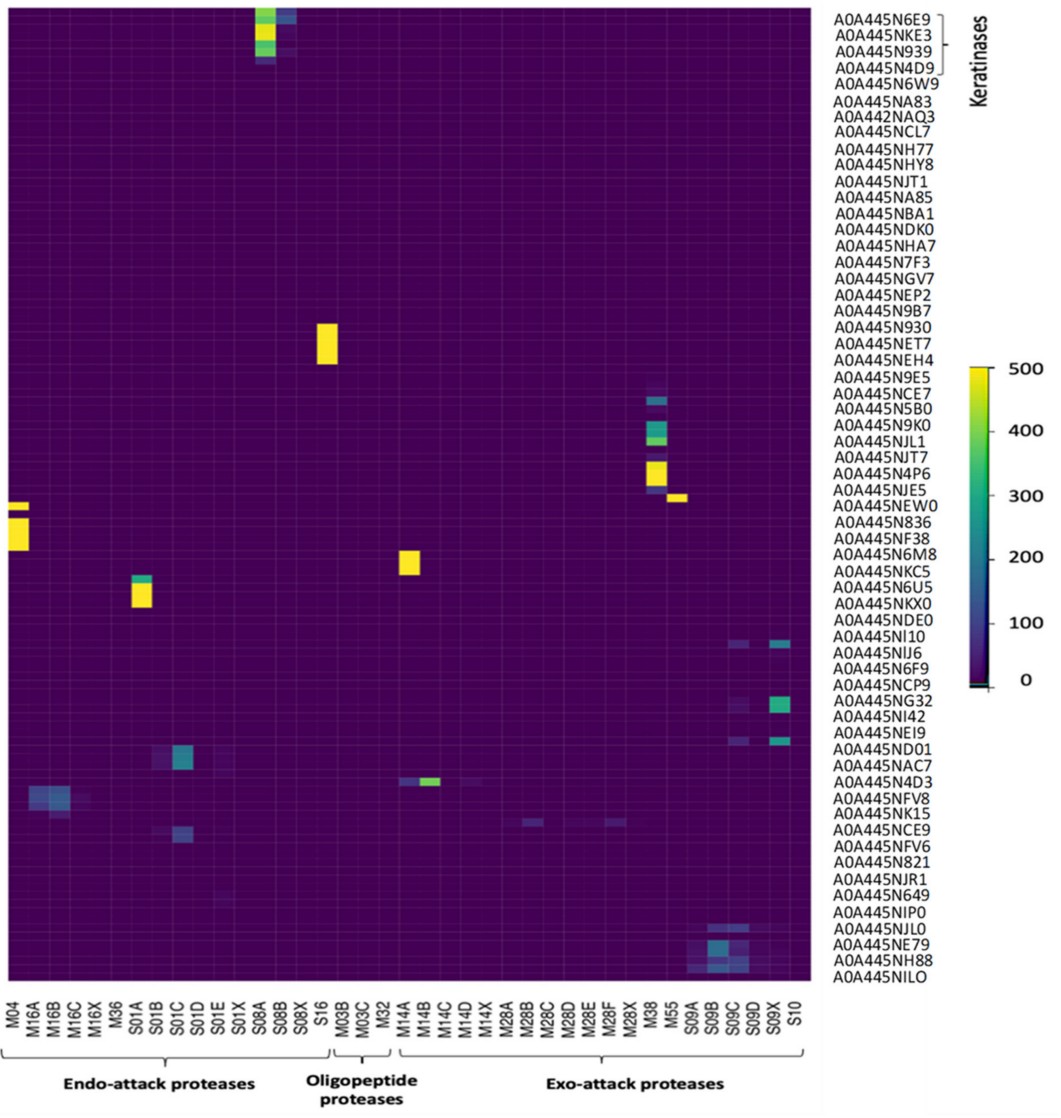

**Figure 6.** Detected proteases of *S. netropsis*.

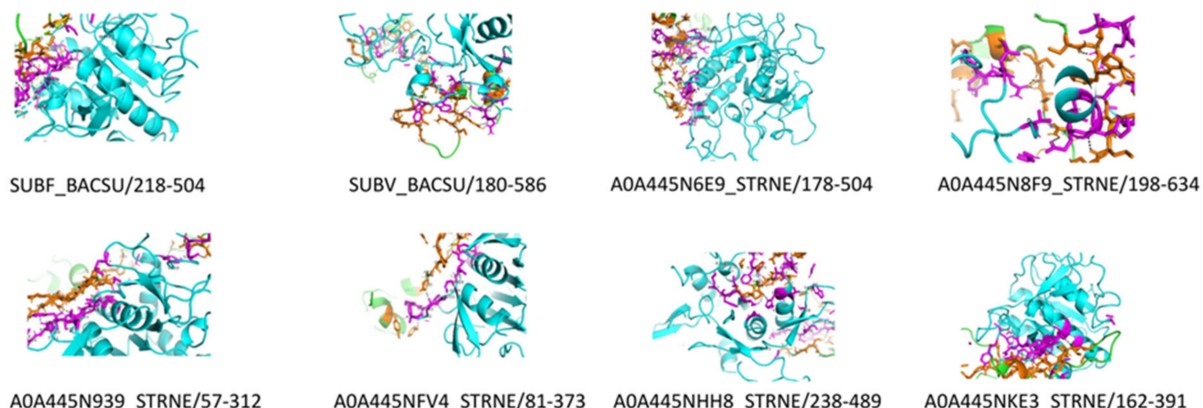

**Figure 7.** Molecular docking complexes of detected keratinases in both strains (*B. subtilis* strain 168, and *S. netropsis*) and beta-keratin (Keratinases in cartoon Cyan, beta-keratin in cartoon Green, Interacted residues of keratinases in stick Magenta, Interacted residues of beta-keratin in stick orange, probable interaction in dashed black).

## 4. Conclusions

The bioprocessing of waste into useful products is an important strategy for sustainable development and the exploitation of renewable resources. Thus, the potential demonstrated by *S. netropsis* A-ICA and *B. subtilis* ALICA are suggestive of the great production of keratinases that bio-convert chicken feathers into soluble proteins and other related products. A neutral to weakly alkaline initial pH medium, 1% (*w/v*) of the substrate (chicken feather), and the mesophilic temperature (25 and 30 °C) were the process conditions that exhibited the optimal keratinase production. The ability of *S. netropsis* A-ICA and *B. subtilis* ALICA to effectively use chicken feathers as a sole carbon and nitrogen source indicates that a variety of enzymes are synthesized to efficiently decompose the complex pertinacious polymer. The high number of total proteins and the high rate of featherweight loss in the fermentation broth are strong indicators of feather degradation by both studied strains. Furthermore, this study exhibited that feather protein hydrolysate possessed an antioxidant potential. Further studies should be conducted to assess the application of these feather hydrolysates as an animal feed additive in vivo. Moreover, bioinformatics analysis provides a comprehensive survey for proteases of *B. subtilis* (strain 168) and *S. netropsis*, the active sites of their keratinases, and possible interaction with beta-keratin. As a result, these hot spots at beta-keratin and keratinases could be a fertile field for molecular biologists to maximize the keratinolytic process, hence accelerating the feather industry wheel.

**Author Contributions:** Conceptualization, A.A., V.M.-T., D.G.-M. and A.F.R.; methodology, A.A., C.C.-D., O.T.-C. and V.M.-T.; supervision, A.F.R., C.M.-C., A.A. and O.G.-J.; writing—original draft preparation, A.A., D.G.-M. and A.F.R.; writing—review and editing, D.G.-M., A.A. and A.F.R. All authors have read and agreed to the published version of the manuscript.

**Funding:** This research received no external funding.

**Informed Consent Statement:** Not applicable.

**Data Availability Statement:** The 16S rRNA gene sequences of identified bacteria were deposited in GenBank with accession number MN535765 and KX137176, respectively.

**Conflicts of Interest:** The authors declare no conflict of interest.

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
