# Peer review of "Keratinases from Streptomyces netropsis and Bacillus subtilis and Their Potential Use in the Chicken Feather Degrading"

_fermentation, doi:10.3390/fermentation9020096_

Round 1

Reviewer 1 Report

The manuscript of Abdelmoteleb et al., assessed the microbial degradation of chicken feathers and the production of keratinases thereof. The authors provide novel insights in the degradation of high keratin feedstocks (in this case chicken feathers) and therefore deserves recognition in this research area. The article is well written (some typos are found and mentioned below) and the applied methodology merits publication in Fermentation. After reading the manuscript I have some major and minor comments which ought to be addressed before publication.

-         - Please mind significant numbers in the text and be consistent with their use.

-         - Please write abbreviations in full when first mentioning them in the text. This will facilitate reading.

-         - Suggestion (Line 60): use “energy-intensive” instead of “energy-exhausted”.

-       -   Suggestion (Line 62) use “ammonia and sulfur dioxide” or “nitrogen and sulfur”.

-        - Typo line 86, remove “finally”.

-         - Capital letter in Line 89 (Keratinous) is not needed.

-         - Please be consistent in naming micro-organisms, sometimes the names appear in italic sometimes not. Please adjust throughout the text.

-        - Please mind the subscript in Line 129.

-        -  Use a spacing between number and unit. Be consistent and adjust throughout the text.

-         - Please introduce a paragraph in Materials and methods that lists all used chemicals.

-        -  Did the authors experimentally determine the exact composition of the feathers (elemental + (bio)-chemical?

-        -  Did the authors considering elemental analysis (N + S) to monitor the degradation/solubilization of feather compounds (proteins + disulfide bonds)?

-        -  Line 152: mind the subscript

-      -    Section 2.4, please add the equipment (+ supplier) that is used to measure absorbance. Equipment is mentioned later on in the manuscript but should be addressed first time it absorbance is mentioned in the manuscript.

-         - Line 167-168: which filter paper is used? Add supplier + pore size.

-         - Point is missing at the end of Line 222.

-        -  Line 237: remove capital letter in “Keratin”. Also in 250 and 336 “Keratinase activity”.

-         - Typo in Line 272 “microorgnains”.

-       -  Please increase font size of Fig. 5 to facilitate reading.

-       -   Currently there is no reference of Fermentation in the reference list. It is advised to a reference(s) of Fermentation to show that it fits into the scope of the journal. Also please have a look at https://doi.org/10.3390/ijerph191710858 , recently published in Environmental Research and Public Health.

Author Response

Response to Reviewer 1 Comments

  1. Please mind significant numbers in the text and be consistent with their use.

Response: The reviewer's suggestion has been considered and incorporated in the text (please see changes in the manuscript)

  1. Please write abbreviations in full when first mentioning them in the text. This will facilitate reading.

Response: The reviewer's suggestion has been considered and incorporated in the text (please see changes in the manuscript)

3.Suggestion (Line 60): use “energy-intensive” instead of “energy-exhausted”.

Response: The changes were realized, please line 60

  1. Suggestion (Line 62) use “ammonia and sulfur dioxide” or “nitrogen and sulfur”.

Response: The changes were realized by ammonia and sulfur dioxide, please see line 62

  1. Typo line 86, remove “finally”.

Response: The changes were realized, please see line 86 -         

  1. Capital letter in Line 89 (Keratinous) is not needed.

Response: The changes were realized, please see line 89         

  1. Please be consistent in naming micro-organisms, sometimes the names appear in italic sometimes not. Please adjust throughout the text.

Response: The reviewer's suggestion has been considered and incorporated in the text (please see changes in the manuscript)

  1. Please mind the subscript in Line 129.

Response: The changes were realized, please see line 129         

  1. Use a spacing between number and unit. Be consistent and adjust throughout the text.

Response: The reviewer's suggestion has been considered and incorporated in the text (please see changes in the manuscript)

  1. Please introduce a paragraph in Materials and methods that lists all used chemicals.

Response: The changes were realized, please see  new Materials and methods section (please see line 146 to 149)

  1.  Did the authors experimentally determine the exact composition of the feathers (elemental + (bio)-chemical?

Response: The reviewer's comment is very accurate. However, in the present study the biochemical composition of the feathers was not determined. Because our objective in the present study, was evaluate the keratinolytic potentials of two native microorganisms   strains using chicken feathers as a sole carbon and nitrogen source.  But in future work, it could be a variable to consider based on the results obtained in our study.  

12.Did the authors considering elemental analysis (N + S) to monitor the degradation/solubilization of feather compounds (proteins + disulfide bonds)?

Response: The reviewer's comment is very accurate. But in this study we not realized elemental analysis to monitor the degradation/solubilization of feather compounds. Our study was in focus to evaluate the keratinase activity in the feather degradation and the analysis of the concentration of soluble protein in the feather hydrolysate. Followed of analysis of disulphide bond reductase activity. However, in future studies can be considered realized the elemental analysis.

13.Line 152: mind the subscript

Response: The changes were realized, please see line 152

  1. Section 2.4, please add the equipment (+ supplier) that is used to measure absorbance. Equipment is mentioned later on in the manuscript but should be addressed first time it absorbance is mentioned in the manuscript.

Response: The changes were realized, please see line 207

  1. Line 167-168: which filter paper is used? Add supplier + pore size.

Response: The reviewer's suggestion has been considered and incorporated in the text (please see changes in the line 221)

16.Point is missing at the end of Line 222.

Response: The changes were realized

  1. Line 237: remove capital letter in “Keratin”. Also in 250 and 336 “Keratinase activity”.

Response: The changes were realized please see line 237, 250 and 336.

  1. Typo in Line 272 “microorgnains”.

Response: The changes were realized please see line 272

  1. Please increase font size of Fig. 5 to facilitate reading.

Response: The reviewer's suggestion has been considered and incorporated two new figures please see figure 5 and figure 6

  1. Currently there is no reference of Fermentation in the reference list. It is advised to a reference(s) of Fermentation to show that it fits into the scope of the journal. Also please have a look at https://doi.org/10.3390/ijerph191710858 , recently published in Environmental Research and Public Health.

Response: The reviewer's suggestion has been considered and incorporated two new references , please see references numbers 30 and 35.

  1. Huang, H.-J.; Weng, B.-C.; Lee, Y.-S.; Lin, C.-Y.; Hsuuw, Y.-D.; Chen, K.-L. The Effects of Two-Stage Fermented Feather Meal-Soybean Meal Product on Growth Performance, Blood Biochemistry, and Immunity of Nursery Pigs. Fermentation20228, 634. https://doi.org/10.3390/fermentation8110634
  2. Możejko, M.; Bohacz, J. Optimization of Conditions for Feather Waste Biodegradation by Geophilic Trichophyton ajelloiFungal Strains towards Further Agricultural Use. Int. J. Environ. Res. Public Health202219, 10858. https://doi.org/10.3390/ijerph191710858
  1. Możejko, M.; Bohacz, J. Optimization of Conditions for Feather Waste Biodegradation by Geophilic Trichophyton ajelloiFungal Strains towards Further Agricultural Use. Int. J. Environ. Res. Public Health202219, 10858. https://doi.org/10.3390/ijerph191710858

Reviewer 2 Report

1. In the title, the first letter should be in capital

2. All scientific names should be in italics.

3.  Line 271, spell of microorganism should be checked

4. In fig 3 A, the protein concentration is being increased up to 9th days and there is no decline trend, so might be on 10th or 11th day the protein concentration become more increased, so authors must consider this.

5. In Fig. 5 the writings mentioned in x-axis and z-axis is difficult to read.

6.  How much inoculum size was used for this study? this also need to be optimized.

Author Response

Response to Reviewer 2 Comments

1.In the title, the first letter should be in capital

      Response: The reviewer's suggestion has been considered and incorporated please see title

2. All scientific names should be in italics.

Response: The reviewer's suggestion has been considered and incorporated in the text (please see changes in the manuscript)

3.  Line 271, spell of microorganism should be checked

Response: The reviewer's suggestion has been considered, please see changes in line 350

4. In fig 3 A, the protein concentration is being increased up to 9th days and there is no decline trend, so might be on 10th or 11th day the protein concentration become more increased, so authors must consider this.

Response: The reviewer's suggestion has been considered, please see explanation in line 397 to 402

5. In Fig. 5 the writings mentioned in x-axis and z-axis is difficult to read.

Response: The reviewer's suggestion has been considered and incorporated two new figures please see figure 5 and figure 6 with large numbers.

6.  How much inoculum size was used for this study? this also need to be optimized.

Response: The reviewer's suggestion has been considered, please see 2.4. Optimization of culture conditions and quantitative assay of keratinase activity section (please see lines 159 to 160).

Round 2

Reviewer 1 Report

The authors have revised the manuscript accordingly and therefore, the reviewer accepts this paper for publication.